# Orange Allergy Beyond LTP: IgE Recognition of Germin-like Proteins in Citrus Fruits

**DOI:** 10.3390/cimb47080621

**Published:** 2025-08-05

**Authors:** M. Soledad Zamarro Parra, Montserrat Martínez-Gomaríz, Alan Hernández, Javier Alcover, Isabel Dobski, David Rodríguez, Ricardo Palacios, Antonio Carbonell

**Affiliations:** 1Allergy Section, Hospital General Universitario Reina Sofía de Murcia, 30003 Murcia, Spain; antonio.carbonell@carm.es; 2DIATER Laboratories, 28918 Leganés, Spain; m.martinez@diater.com (M.M.-G.); a.hernandez@diater.com (A.H.); alcover@diater.com (J.A.); idobski@diater.com (I.D.); d.rodriguez@diater.com (D.R.); r.palacios@diater.com (R.P.)

**Keywords:** food allergy, orange, citrus fruits, germin, IgE, LTP-independent allergy

## Abstract

Orange allergy is estimated to account for up to 3–4% of food allergies. Major allergens identified in orange (*Citrus sinensis*) include Cit s 1 (germin-like protein) and Cit s 2 (profilin), while Cit s 3 (non-specific lipid transfer protein, nsLTP) and Cit s 7 (gibberellin-regulated protein) have also been described. The objective of this study was to investigate the presence and IgE-binding capacity of germin-like proteins in citrus fruits other than oranges. We describe five patients with immediate allergic reactions after orange ingestion. All patients underwent skin prick tests (SPT) to aeroallergens and common food allergens, prick-by-prick testing with orange, lemon, and mandarin (pulp, peel, seeds), total IgE, specific IgE (sIgE), anaphylaxis scoring (oFASS), and the Food Allergy Quality of Life Questionnaire (FAQLQ-AF). Protein extracts from peel and pulp of orange, lemon, and mandarin were analyzed by Bradford assay, SDS-PAGE, and IgE immunoblotting using patient sera. Selected bands were identified by peptide mass fingerprinting. A 23 kDa band was recognized by all five patients in orange (pulp and peel), lemon (peel), and mandarin (peel). This band was consistent with Cit s 1, a germin-like protein already annotated in the IUIS allergen database for orange but not for lemon or mandarin. Peptide fingerprinting confirmed the germin-like identity of the 23 kDa bands in all three citrus species. Germin-like proteins of approximately 23 kDa were identified as IgE-binding components in peel extracts of orange, lemon, and mandarin, and in orange pulp. These findings suggest a potential shared allergen across citrus species that may contribute to allergic reactions independent of LTP sensitization.

## 1. Introduction

Food allergy is a major public health problem, with prevalence rates increasing over the past twenty to thirty years. According to community-based EuroPrevall surveys, the prevalence of self-reported among adults in Europe ranges from 2% to 37% [1].

Nearly half of adults with food allergies have experienced at least one adult-onset reaction, and 38% have reported at least one food allergy-related visit to an emergency department during their lifetime [2]. There are many challenges in estimating food allergy prevalence [2,3]. A recent systematic review reported a lifetime prevalence of any food allergy diagnosed by a physician of 9.3% in children and 5% in adults [4].

Citrus fruits, particularly oranges (*Citrus sinensis*), belong to the family *Rutaceae*. Orange consumption has been linked to allergic reactions, including oral allergy symptoms such as itching and swelling of the lips, tongue, and throat [5,6].

Oranges are consumed worldwide. Despite their widespread use in food and flavoring, reports of allergies to orange, lemon, or mandarin are relatively rare, with orange allergies accounting for only 4.4% of all food allergies [2], reported by Lyons in 2020.

Four primary allergens have been described in orange, labeled Cit s 1 through Cit s 4, with the major allergens being Cit s 1 (a germin-like protein) and Cit s 2 (profilin) [7]. Cit s 1, a 24 kDa glycoprotein [7], is also found in lemon peel [8]. Its biological role is still under investigation, but it is considered one of the main orange allergens [7]. Cit s 2, a 14 kDa profilin, is another major orange allergen [7,9]. Like other profilins, its biological activity is related to the structural organization of actin filaments and germination [10]. Cit s 3, a 9.46 kDa lipid transfer protein (LTP) type 1, belongs to the LTP family of panallergens. Despite its LTP classification, it is considered a minor allergen with a prevalence of 34% [11]. Like most LTP, it is most abundant in the peel but is also present in the pulp [12]. Cit s 7, a 6.951 kDa gibberellin-regulated protein [13], was tested for allergenicity in a group of 14 patients with orange allergies, with 85.7% of patients [13] testing positive for it through ELISA, Prick, or basophil activation tests. Cross-reactivity between orange and lemon allergens and LTP of peach (Pru p 3) has also been described [12].

Regarding lemon (*Citrus limon*), one allergen has been described: Cit l 3, a 9.6 kDa lipid-transfer protein type 1. Among 27 sera analyzed, specific IgE to the purified allergens was found in 54% for nCit l 3, 48% for nCit s 3, 46% for rCit s 3, and in 37% for rPru p 3. Members of the LTP allergen family are involved in allergy to oranges, displaying positive in vitro and in vivo reactions seen in 30–50% of patients studied. Both orange and lemon allergens show cross-reactivity with the major peach allergen, Pru p 3 [12].

In mandarin (*Citrus reticulata*), the main allergen identified is Cit r 3, a 9 kDa lipid-transfer protein type 1, found in a patient with mandarin-induced anaphylaxis [14]. This case, which occurred in Northern and Central Europe, was unusual, as fruit-induced anaphylaxis is rare in this region. It also highlighted the importance of sensitization to LTP and the consequent predisposition to severe allergic reactions [14]. 

Germins are cell wall proteins expressed during the early stages of germination in response to physical or chemical stress [15].

We describe five patients with clinical reactions to orange but no evidence of LTP sensitization, suggesting a role for germin-like proteins in citrus allergy in the Mediterranean area. Our aim was not to identify novel allergens, but to investigate the role of germin-like proteins in orange, lemon, and mandarin, as a possible cause of allergy beyond lipid transfer proteins (LTPs).

## 2. Materials and Methods

We report five cases of patients—four women and one man—aged between 31 and 44 years, who first attended our allergy clinic with suspected orange allergy. The characteristics of the patients and the diagnostic procedures performed are detailed below.

All five patients developed immediate allergic reactions following orange ingestion, presenting with symptoms such as sneezing, nasal congestion, erythema over the chest, and shortness of breath, along with generalized urticaria and facial angioedema. The reactions were classified as ordinal Food Allergy Severity Score (oFASS) grade 4, indicating severe food allergy [1]. 

### 2.1. Skin Prick Test

Skin prick tests (SPT) were performed using commercial extracts of common aeroallergens, including pollen, house dust mite, molds, animal dander, and panallergens, such as peach non-specific lipid transfer proteins (nsLTPs) (Pru p 3) and pollen profilins.

SPTs were conducted for the following allergens:Pollens: *Phleum pratense*, *Cynodon dactylon*, *Olea europaea*, *Cupressus arizonica*, *Platanus acerifolia*, *Artemisia vulgaris*, *Plantago lanceolata*, and *Parietaria judaica*.Animal dander: dog and cat.Molds: *Alternaria alternata*, *Aspergillus fumigatus*.Panallergens: Peach nsLTP (Pru p 3, Diater^®^; 30 µg/mL) and pollen profilin (Pho d 2).

Additionally, SPTs for food allergens were performed using commercial extracts for cocoa, hazelnut, peanut, almond, sunflower seed, pistachio, walnut, peach, apple, pear, kiwi, banana, melon, strawberry, pineapple, tomato, celery, and paprika (Diater^®^ Laboratories, Leganés, Madrid, Spain). 

### 2.2. Prick-by-Prick Test

The prick-by-prick test was performed using fresh samples of food.

Orange (*Citrus sinensis*): peel and pulpLemon (*Citrus limon*): peel and pulpMandarin (*Citrus reticulata*): peel and pulpCitrus seeds.

### 2.3. Anaphylaxis Score and Immunological Tests

The anaphylaxis severity score was assessed using the oFASS-5 classification. Additionally, the following immunological tests were performed:Total IgE (kU/L)Serum-specific IgE (sIgE) to nsLTP (Pru p 3, Pru p 7) (Thermo Fisher Scientific Inc., Phadia, AB, Uppsala, Sweden)The ALEX multiplex allergy test (MacroArray Diagnostics GmbH, Vienna, Austria) was used to assess specific IgE sensitizations. This test allows simultaneous measurement of allergen extracts and molecular components [16]Food Allergy Quality of Life Questionnaire Adult Form (FAQLQ-AF) [3,4].

### 2.4. Preparation of Allergenic Extracts

Allergenic extracts were prepared from orange (*Citrus sinensis*) peel and pulp, lemon (*Citrus limon*) peel and pulp, and mandarin (*Citrus reticulata*) peel and pulp. Samples were washed with deionized water, then weighed and homogenized in a beaker with Björksten Buffer (NaH_2_PO_4_ 0.14% (*w*/*v*), NaH_2_PO_4_ × 2 H_2_O 0.18% (*w*/*v*), polyvinylpolypyrrolidone 2% (*w*/*v*), EDTA 0.07% (*w*/*v*), and diethyl dithiocarbonate 0.07% in deionized water). The mixture was blended until a homogeneous suspension was obtained under gentle mixing.

The protein concentration was standardized to 10 mg/mL of buffer. This buffer was used for the extraction and stabilization of proteins in fruit and vegetables. The pH of the extracts was adjusted to 7.7–8.5 by adding 1M NaOH. This was followed by extraction under gentle mixing in a cold chamber for protein extraction and stabilization in fruit and vegetable samples. The pH was adjusted to 7.7–8.5 using 1M NaOH, followed by extraction under gentle mixing at 4 °C for 2 h using a magnetic stirrer (Thermo Fisher Scientific Inc. Phadia, AB, Uppsala, Sweden). The extract was then centrifuged at 9000 rpm for 30 min at 4 °C.

The supernatant was filtered through a series of clarification filters with pore sizes of 25–30 µm, 10–13 µm, and 2 µm, sequentially removing impurities, sediments, and suspended particles.

### 2.5. Protein Content

The total protein concentration of the allergenic extracts was determined using the Bradford method [5]. The assay was performed with the Bio-Rad Protein Assay reagent diluted 1:5 in ultrapure water. A standard curve was generated using a bovine serum albumin (BSA) stock solution at 1 mg/mL, followed by serial dilutions. A blank (negative control) containing only buffer was included.

For each sample, 900 µL of Bradford reagent was added to the extract, vortexed briefly, and incubated for 5 min at room temperature. Subsequently, 200 µL of each reaction mixture was loaded in triplicate into a 96-well microplate. Absorbance was measured at 595 nm using a Power Wave XS2 spectrophotometer (BioTek Instruments, Winooski, VT, USA) and Gen5™ software 2.09.

Protein concentrations were calculated by interpolating the absorbance values on the standard curve. A correlation coefficient (R^2^) ≥ 0.99 was considered acceptable for curve validity.

### 2.6. SDS-PAGE and Immunoblotting

After determining the protein concentration in the allergenic extracts, the dilution formula (C1 × V1 = C2 × V2) was used to bring the samples to 0.05 mg/mL under reducing conditions (optimal concentration for SDS-PAGE and immunoblotting [Western blot]).

For reducing conditions, the samples were mixed with 5× Sample Buffer, composed of the following:-0.5 M Tris-HCl (pH 6.8) at 12% (*v*/*v*),-Glycerol at 25% (*v*/*v*),-SDS at 10% (*w*/*v*) in a 20% concentration (*v*/*v*),-2-mercaptoethanol at 5% (*v*/*v*),-Bromophenol blue at 0.5% (*w*/*v*) in a 20% concentration (*v*/*v*),-Deionized water at 18% (*v*/*v*).

Samples were denatured by heating at 99 °C for 10 min.

#### 2.6.1. SDS-PAGE Analysis

Denatured samples were loaded at 0.5ac µg per lane onto 15% acrylamide/bisacrylamide gels under denaturing conditions. Electrophoresis was conducted using the PowerPac 300 BASIC (Bio-Rad, Hercules, CA, USA) following the Laemmli method [6] at:-200 V, 400 mA for 45 min,-Electrophoretic cuvette filled with 10 × Tris/Glycine/SDS electrophoresis buffer (Bio-Rad).

For molecular weight determination, Precision Plus Dual Color Protein Standards (Bio-Rad) (ranging from 250 kDa to <10 kDa) were used at a 1:5 dilution ratio with 1× sample buffer.

Proteins were visualized using the Pierce^TM^ Silver Stain Kit (Thermo Scientific), following the manufacturer’s protocol. Molecular weights were analyzed using a Bio-Rad transilluminator and Image Lab 5.2.1 software.

#### 2.6.2. Western Blotting

To assess sIgE sensitization to citrus allergens, extracts were subjected to Western blot analysis. Proteins were separated by SDS-PAGE (15% gels, 0.5 per lane) under reducing conditions, using the PowerPac 300 BASIC system (Bio-Rad) with the Laemmli method [1] at 200 V, 400 mA, for 45 min.

Following electrophoresis, proteins were transferred onto a polyvinylidene fluoride (PVDF) membrane (Trans-Blot Turbo Transfer Pack, Bio-Rad) using the Trans-Blot Turbo system (Bio-Rad).

The membrane was blocked for 1 h at room temperature (RT) with phosphate-buffered saline (PBS, pH 7.4) containing 0.5% Tween and 3% milk powder.

For IgE detection, the membrane was incubated overnight with patient serum (primary antibody) at a 1:5 dilution in PBS-Tween (0.5%) and 3% milk powder. Following incubation, the membrane underwent five washes with 0.5% PBS-Tween. 

The primary antibody was detected using horseradish peroxidase (HRP)-conjugated mouse anti-human IgE Fc (SouthernBiotech, Birmingham, AL, USA), followed by chemiluminescence detection with Western Lightning Plus-ECL (PerkinElmer, Inc., Waltham, MA, USA), according to the manufacturer’s instructions. 

### 2.7. Protein MALDI-TOF/TOF Identification

To identify IgE-binding proteins from lemon, mandarin, and orange, protein bands of interest were excised from the SDS-PAGE gel and submitted to the Proteomics Department of the Faculty of Pharmacy of Universidad Complutense de Madrid (UCM) [Madrid Complutense University] for analysis via peptide mass fingerprinting (MALDI-TOF/TOF). 

In this method, the target protein undergoes enzymatic digestion, typically using trypsin, which hydrolyzes the protein into peptides at specific cleavage sites. The resulting peptide fragments are then analyzed using a mass spectrometer equipped with an appropriate ionization source, such as Matrix-Assisted Laser Desorption/Ionization Time-of-Flight (MALDI-TOF) or Electrospray Ionization Time-of-Flight (ESI-TOF). The absolute peptide masses obtained are used to generate a peptide mass fingerprint enabling protein identification.

## 3. Results

### 3.1. Skin Prick Test. Prick-by-Prick Tests *(See Table 1)*

Patient 1:Skin prick test (SPT to commercial aeroallergen extracts): Negative.SPT to commercial food extracts: Negative.Prick-by-prick test:
oOrange: positive for pulp and peel (5 mm major diameter papule).oLemon: positive for pulp (3 mm major diameter papule), negative for peel.oMandarin: positive for pulp (3 mm major diameter papule), negative for peel.The patient tolerated an oral lemon and mandarin challenge. For more information see Table 2.

Patient 2:SPT to aeroallergens: positive for cat, dog, *Phleum pratense*, *Cynodon dactylon*, *Olea europaea*, *Salsola kali*, *Chenopodium album*.SPT to foods: positive to peach nsLTP (Pru p 3), hazelnut, almond, peanut, and pistachio nut.Prick-by-prick:oOrange: positive for pulp (7 mm major diameter papule) and peel (8 mm).oLemon: positive for pulp (9 mm major diameter papule) and peel (10 mm papule).oMandarin: positive for pulp (9 mm major diameter papule) and peel (10 mm papule).oLemon seed: positive (7 mm major diameter papule).oMandarin seed: negative. Orange seed is not tested.


Patient 3:SPT to aeroallergens: positive for *Olea europaea*, *Parietaria judaica*, *Artemisia vulgaris*, *Salsola kali*, and *Chenopodium album*.SPT to foods: positive for peach nsLTP (Pru p 3), corn flour, soy flour, hazelnut, walnut, sunflower seeds, and tomato.Prick-by-prick:oOrange: positive for pulp and peel. Diameter is not available. oLemon and mandarin: not tested. 


Patient 4:SPT to commercial aeroallergen extracts: Negative.SPT to food: positive for hazelnut, almond, peanut, walnut, pistachio, apple, strawberry, and peach nsLTP (Pru p 3).Prick-by-prick:oOrange: positive for pulp (6 mm) and peel (6 mm). oLemon and mandarin: not tested.

Patient 5:SPT to commercial aeroallergen extracts: positive for *Alternaria alternata*, *Artemisia vulgaris*, *Chenopodium album*, *Salsola kali*, *Phleum pratense*, *Cupressus arizonica*, *Olea europaea*, and *profilin* (Pho p 2).SPT to foods: positive for peach, watermelon, and melon.Prick-by-prick:
oPositive for orange pulp (5 mm) and peel (5 mm).oPositive for apple pulp and peel.oNegative for lemon and mandarin. The patient tolerated an oral lemon and mandarin challenge.
Orange (pulp and peel): positive in all five patients.Oral food challenges with lemon and mandarin were performed in Patients 1 and 5, with no adverse reactions observed. In the remaining cases, no clinical history of reactions to these fruits was reported, and they all tolerated in juice; they refused to eat it whole (see Table 3).

**Table 1 cimb-47-00621-t001:** Number of positive SPT results (for aeroallergens and foods) and prick-by-prick results (major diameter in millimeters) for orange, lemon, and mandarin pulp and peel. Zero is for negative. POS is for papule diameter > 3mm. NEG is for papule < 3 mm.

					No Positives							Positives			
		Prick								Prick Prick (Dia/mm)					
	Inhalants					Foods				Orange		Lemon		Mandarin	
Patient	Dander	Tree	Grass	Weed	Mould	Fruit/Veg	Wheat	Peanut	Tree Nut Seed	Pulp	Peel	Pulp	Peel	Pulp	Peel
1	0	0	0	0	0	0	0	0	0	5	5	3	0	3	0
2	2	0	0	0	0	0	0	6	0	7	8	9	10	9	10
3	0	0	0	4	0	0	0	2	1	POS	POS	ND	ND	ND	ND
4	0	0	0	0	0	0	0	4	1	POS	POS	ND	ND	ND	ND
5	0	3	0	3	0	3	0	0	0	POS	POS	NEG	NEG	NEG	NEG

### 3.2. Molecular Diagnosis and Symptom Scoring

Total IgE and LTP-specific IgE levels varied among patients (see Table 2).Only Patient 1 showed positive IgE reactivity to all tested recombinant allergens.The remaining four patients recognized only 4, 4, 1, and 0 recombinant allergens, respectively.Food Allergy Quality of Life Questionnaire (FAQLQ) scores ranged from 99 to 166 points.

**Table 2 cimb-47-00621-t002:** Orange sensitization, molecular allergens (kU/L), rhinitis/asthma presence, symptom scores (FAQLQ), and tryptase levels. nsLTP allergens are highlighted in bold.

Patient	IgE **Pru p 3**	IgE Pru p 7	Total IgE (KU/L)	Prick Test to Orange (Major Diameter mm)	IgE *Citrus sinensis*	Rhinitis Symptoms	Spring Asthma Symptoms	Sensitization to Pollen	**Ara h 9**	**Art v 3**	**Ole e 7**	Art v 1	Ole e 1	Che e 1	Sal k 1	Cup a 1	FAQLQ	Basal Tryptase
1	4.3	0	57	6	3.56	Yes	No	No	0	0	0	0	-	0	-	-	79	4.8
2	0.02	0	518	8	3.54	Yes	No	Yes	0	0	0	0	1.5	0	21	18.6	166	3.79
3	0.7	0	336	7	2.12	Yes	Yes	Yes	2	0.5	15	1	13	32	47	0.6	150	4.5
4	3.84	0	17.2	6	0.15	Yes	No	Yes	-	-	-	-	-	15	12	-	122	5.6
5	0	0	52	5	2.13	Yes	No	Yes	0	0	0	2	12	0	7	0	140	3.6

**Table 3 cimb-47-00621-t003:** Clinical characteristics of the first allergic episode.

Patient	Gender	Age	Profession	First Reaction to Food Reported from Patients	First Food Involved	Anaphylaxis	Previous Reactions to Other Foods	Tolerance to Peach Juice: Yes/No	Orange Episode	Adrenaline Needed	oFASS	Cofactors
1	Male	33	Nuclear Medicine Technician	2013	Orange	Yes	Anaphylaxis to nuts	Yes	Anaphylaxis after eating oranges	Yes	4	No
2	Woman	43	Street vendor	2020	Orange	Yes	Palmar pruritus, touching nuts, and anaphylaxis mediated by NSAIDs due to almonds	Yes	Anaphylaxis after eating oranges	Yes	4	No
3	Woman	34	Worker in an orange juice factory	2020	Peach	No	Oral allergy syndrome to peach	No	Cough/Asthma to inhale orange juice	No	2	No
4	Woman	31	Pharmacy Technician	2007	Peach	No	Urticaria after eating a peach with the peel	No	Urticaria and facial angioedema after eating an orange	No	2	No
5	Woman	33	Housewife	2011	Pomegranate	No	Oral allergy syndrome to pomegranate	Yes	Oral allergy syndrome to orange	No	1	No

### 3.3. Characterization of Relevant Proteins 

After extract preparation, protein concentrations (mg/mL) were determined as follows:Orange peel: 0.17Orange pulp: 0.06Lemon peel: 0.06Lemon pulp: 0.03Mandarin peel: 0.12Mandarin pulp: 0.04.

#### Electrophoretic Analysis

Figure 1 presents SDS-PAGE profiles of citrus fruit peel and pulp extracts:Orange (lane 1–2):oBands corresponding to profilin (Cit s 2, 14 kDa) are detected in both peel and pulp. oA protein approximately 23 kDa, possibly corresponding to germin, is also present. oNo bands below 10 kDa (e.g., Cit s 3, nsLTP, 9.46 kDa; Cit s 7, gibberellin, 8 kDa) were detected, likely due to low protein concentration. However, their presence cannot be ruled out. oAdditional undocumented bands not listed by the International Union of Immunological Societies (IUIS) were also observed. 
Lemon (lanes 3–4):oA faint band ~10 kDa, potential Cit s 3 (an nsLTP identified at 9.6 kDa), was detected in both peel and pulp. oAdditional proteins with molecular weights ranging from 15 to 75 kDa were also observed.
Mandarin (lanes 5–6):oA faint band slightly above 10 kDa was detected, which may correspond to Cit r 3, an nsLTP at 9 kDa. oThe number of detected proteins was higher in the peel extract (lane 5) than in the pulp extract (lane 6). 


The next figure shows SDS-PAGE profiles of citrus extracts under non-reducing conditions (without 2-mercaptoethanol).

**Figure 1 cimb-47-00621-f001:**
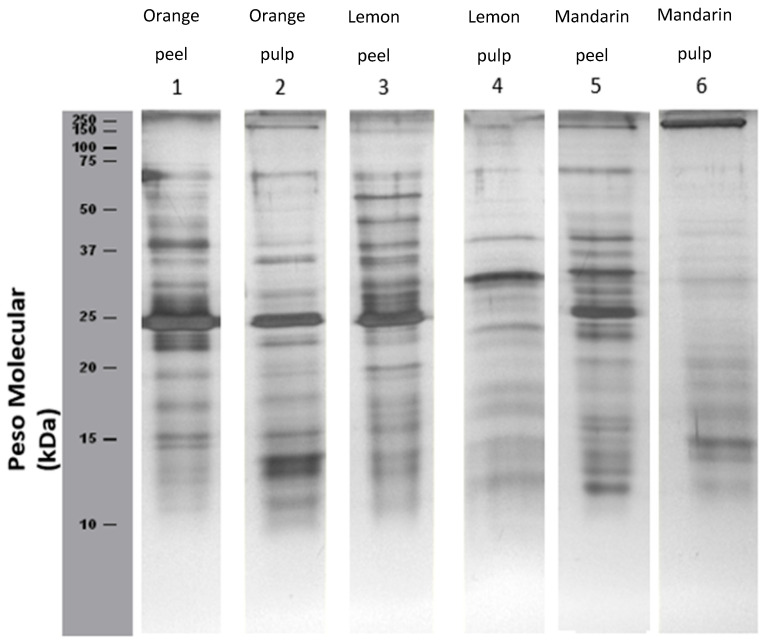
SDS-PAGE with orange, lemon, and mandarin extracts. Lane 1, orange peel-extract; lane 2, orange pulp-extract; lane 3, lemon peel-extract; lane 4, lemon pulp-extract; lane 5, mandarin peel-extract; lane 6, mandarin pulp-extract.

Once the extracts were characterized, they were tested against the sera of the five selected patients (Figure 2).

Strikingly, all patients recognized a protein of around 23 kDa present in orange peel and pulp, lemon peel, and mandarin peel. In contrast, recognition of lemon and mandarin pulp extracts was much lower in most patients, except for patient 3, who strongly recognized a protein at about 30 kDa (above) in the lemon pulp extract.

Also, Patients 1, 2, 3, and 5 recognized a protein of around 63 kDa present in orange peel and pulp, lemon peel, and mandarin peel. This 63 kDa protein band, identified by immunoblotting, was excised and submitted for peptide mass fingerprinting. However, the analysis was inconclusive due to insufficient protein mass in the gel fragment to obtain reliable sequencing data.

### 3.4. Allergen Sequences

To confirm specific allergen binding in Western blotting across all five patients, we selected the 23 kDa band, which was strongly recognized by all of them. This band, detected in orange (pulp and peel), lemon (peel), and mandarin (peel and pulp), has already been annotated in IUIS in the case of *Citrus sinensis* (orange) as a germin (Cit s 1).

To perform peptide fingerprinting, SDS-PAGE (15% and 10%) was carried out for the three citrus fruits. As a positive control, the 23 kDa band from orange—compatible with Cit s 1 (germin) and already annotated in IUIS—was selected. The 23 kDa bands from orange, lemon, and mandarin peel were excised and submitted for peptide fingerprint analysis, yielding the following sequences (Table 4). 

**Table 4 cimb-47-00621-t004:** Sequences of 23 kDa bands are isolated from orange, lemon, and mandarin.

Access	Mass		Score *	Description
Orange	AOAO67CZF3	24,270	80	Germin-like protein
Lemon	AOA2H5QHS8	24,303	70	Germin-like protein
Mandarin	AOA2H5QHS8	24,303	100	Germin-like protein

* Scores above 63 indicate statistically significant values (*p* < 0.05).

The specific values for each of the citrus fruits were as follows:-Orange: Score = 80; E-value: 0.0013; Monoisotopic mass (Mr): 24270 and Calculated pI: 5.76.-Lemon: Score = 70; E-value: 0.012; Monoisotopic mass (Mr): 24303 and Calculated pI: 6.06.-Mandarin: Score = 100; E-value: 1.2 × 10^−5^; Monoisotopic mass (Mr): 24303 and Calculated pI: 6.06.

## 4. Discussion

Although LTP sensitization to orange (Cit s 3) occurs in up to 50% of LTP-allergic patients [12], the clinical tolerance to orange is often higher than that to other LTP-rich foods such as peach, particularly in our Mediterranean population. This is consistent with our findings: Among five patients with confirmed clinical allergy to orange, none showed relevant sensitization to Cit s 3 or cross-reactive nsLTPs (e.g., Pru p 3), and only one had a positive IgE level to Pru p 3 (4.3 kU/L). Notably, four of the five patients had prior tolerance to peach or pomegranate, suggesting a distinct allergenic mechanism.

To date, EAACI guidelines do not endorse orange as a universally safe food in LTP allergy [17]. Although several citrus allergens have been described, and oranges are considered allergenic sources to be taken into account, sensitization to citrus does not usually involve manifestations as obvious as those caused by other foods [12]. In this regard, our case series highlights a relevant diagnostic caveat: All five patients tested positive for orange pulp and peel by prick-by-prick testing but lacked citrus-LTP sensitization, and two underwent oral challenge with lemon and mandarin without adverse reactions, supporting the hypothesis of an alternative IgE-binding target.

Furthermore, while oral challenge-confirmed citrus fruit allergies represent only one-third of suspected cases in larger series [18], all our patients reported reproducible clinical reactions to orange. This suggests that germin-like proteins such as Cit s 1 may account for a small but clinically relevant subset of orange allergies that are independent of LTP, particularly in regions where LTP-based diagnosis dominates clinical algorithms.

To date, the IUIS database lists the following allergens for orange: Cit s 1 (germin, 23 kDa), Cit s 2 (profilin, 14 kDa), Cit s 3 (nsLTP, 9.46 kDa), and Cit s 7 (gibberellin, 8 kDa). However, for lemon and mandarin, only Cit l 3 and Cit r 3, both nsLTP of 9.6 and 9 kDa, respectively, have been annotated.

Regarding citrus fruit extracts in general—and in our study—the protein concentrations obtained were low, with averages of 0.11 mg/mL for peel and 0.04 mg/mL for pulp. These low concentrations resulted in very small working amounts (0.05 mg/mL) for electrophoresis and Western blotting, limiting antigen availability for visualization and recognition. 

The patients studied exhibited comparable sensitization profiles regarding the number and intensity of IgE-binding bands for orange pulp and peel. In contrast, for lemon and mandarin, IgE reactivity was generally more intense in the peel than in the pulp. None of the five patients showed IgE binding to profilin (Cit s 2) or to nsLTPs such as Cit s 3, nor did they recognize the gibberellin-regulated protein band at ~8 kDa (Cit s 7). However, four out of five patients exhibited IgE-binding to protein bands within the 50–75 kDa range. These bands were detected in extracts from both peel and pulp, with the strongest recognition observed in lemon and mandarin peel extracts.

In the case of mandarin peel, four out of five patients showed IgE-binding to two distinct protein bands: one at approximately 32 kDa, which was further analyzed by peptide mass fingerprinting, and another at a slightly lower molecular weight (~27–28 kDa), which remains uncharacterized and may represent a novel candidate allergen. In Patient 4, the signal intensity for both bands was weak but clearly distinguishable from adjacent negative control lanes, suggesting specific IgE reactivity despite lower band intensity.

Overall, no greater reactivity was observed for the peel than for the pulp, ruling out a predominance of LTP in the reactions. In our cohort, the mean wheal diameters in prick-by-prick testing were comparable between pulp and peel for all three citrus fruits—for example, 4.84 mm for orange pulp vs. 5.04 mm for peel, and 4.97 mm for both pulp and peel of mandarin. This absence of greater reactivity to the peel is particularly relevant because non-specific lipid transfer proteins (nsLTPs), such as Cit s 3, are typically more abundant in the peel than in the pulp. Therefore, if nsLTPs were the main allergens involved, one would expect larger wheals to peel extracts. However, the balanced response we observed, together with the absence of specific IgE to Pru p 3 or Cit s 3 in most patients, argues against a predominant role of LTPs. Instead, these findings support the involvement of alternative allergens, such as germin-like proteins, in the pathogenesis of citrus allergy in this subgroup of patients. In most cases, prior reactions to other foods involved nuts and peaches. In these patients, citrus-related symptoms trigger anaphylaxis.

A 23 kDa IgE-reactive band, consistent with a germin-like protein (Cit s 1), was detected in all five patients in both orange pulp and peel, and in the peel of lemon and mandarin. These findings suggest a broader distribution of germin-like proteins across citrus species than previously described in the IUIS allergen database, which only includes Cit s 1 for *Citrus sinensis* (orange).

Previous studies have shown that germin-like proteins such as Cit s 1, as well as profilins like Cit s 2, are heat-stable and retain their allergenicity in processed citrus juices [12]. Therefore, the identification of germin in lemon and mandarin—particularly in patients without LTP sensitization—may represent a clinically relevant allergenic source that has so far been overlooked in diagnostic platforms.

The presence of IgG in the patients’ sera cannot be ruled out. However, the Western blot images obtained (revealed with anti-human IgE) and the detection of specific IgE to *Citrus sinensis* at class 3 (3.5–17.5 kU/L) in two patients and class 2 (0.70–3.5 kU/L) in two others indicate the predominant role of an IgE-mediated response in Patients 1, 2, 3, and 5. Patient 4’s specific IgE to *Citrus sinensis* was 0.15 kU/L, which is within the normal range. However, the 6 mm wheal observed in the prick test suggests an IgE-mediated mechanism. 

## 5. Conclusions

In this study, a 23 kDa IgE-binding band corresponding to a germin-like protein was consistently detected in all five patients, in both pulp and peel of orange, and in the peel of lemon and mandarin. Although this protein (Cit s 1) is already annotated in the IUIS allergen database for *Citrus sinensis*, it has not previously been described for *Citrus limon* or *Citrus reticulata*.

Peptide mass fingerprinting confirmed the identity of these bands as germin-like proteins. Their presence was confirmed in the peel of all three citrus fruits and in the pulp of orange, but not in the pulp of lemon or mandarin.

The detection of germin-like proteins in multiple citrus species suggests a potential source of IgE cross-reactivity that may contribute to clinical symptoms in patients with orange allergy, even in the absence of LTP sensitization. Although cross-reactivity was not formally assessed in this study, our findings support the inclusion of germin-like proteins in future allergen panels and warrant further investigation.

## Figures and Tables

**Figure 2 cimb-47-00621-f002:**
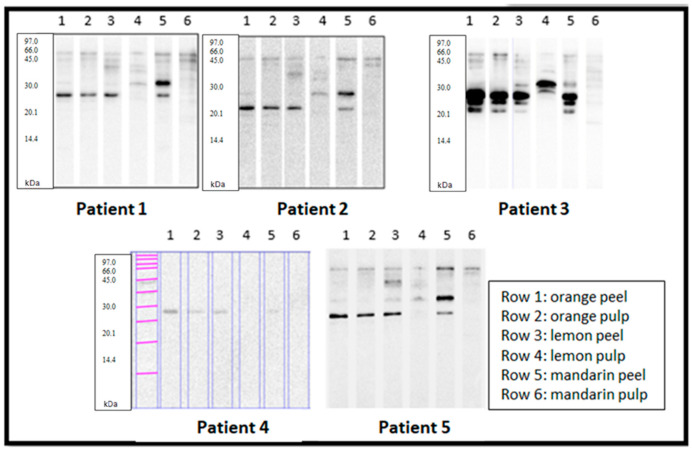
IgE immunoblot after SDS-PAGE with protein extract from orange, lemon, and mandarin (Patients 1 to 5, from left to right and from top to bottom). SDS-PAGE/IgE Western blot: lane 1: orange peel; lane 2: orange pulp; lane 3: lemon peel; lane 4: lemon pulp; lane 5: mandarin peel; lane 6: mandarin pulp.

## Data Availability

The data presented in this study are available on reasonable request from the corresponding author. The data are not publicly available due to privacy and ethical restrictions.

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
