# Peer review of "Orange Allergy Beyond LTP: IgE Recognition of Germin-like Proteins in Citrus Fruits"

_cimb, 2025, doi:10.3390/cimb47080621_

Round 1
Reviewer 1 Report (New Reviewer)
Comments and Suggestions for Authors
I have read with great interest the article entitled Germin: relevant Allergen in lemon, orange and tangerine. This article is an example of a thorough study, well developed, very well explained.
The introduction introduces the topic to be explained very well.
The methods are a compendium of a very complete test from an excellent protein laboratory.
The results are well developed although I miss some photographs of the results of some of the patients' prick tests.
The discussion is very well argued.
Author Response
Dear Reviewers
I hope this letter finds you well.
I have reviewed my manuscript and made several revisions. Below are the changes I have made:
In the title I have used capital letters as in formal english. In formal English, capitalization rules for titles can vary slightly depending on the style guide being followed (e.g., APA, MLA, Chicago). However, a general rule for capitalizing titles is:
- Capitalize the first and last words of the title.
- Capitalize all major words in the title (including nouns, pronouns, verbs, adjectives, and adverbs).
- Do not capitalize articles (a, an, the), coordinating conjunctions (and, but, or), or prepositions unless they are the first or last word in the title.
Based on this rule, your title would be correct as:
"Germin: Relevant Allergen in Lemon, Orange, and Mandarin"
- Flavouring was incorrect; it has been changed to flavoring (correct).
- Analised was incorrect; it has been changed to analyzed (correct).
- In Table 1, I noticed that the following clarification was missing:
- Zero is for negative. POS is for papule diameter > 3mm. NEG is for papule < 3 mm.
- Also in Table 1, I changed the color letters from red to black.
- I have corrected kUA/L to kU/L (the correct form).
- Characterised was incorrect; it has been changed to characterized (correct with “z”).
- 24270 has been changed to 24,270 (the comma was added for clarity, as it represents twenty-four thousand, two hundred seventy).
- Regarding citrus fruit extracts, I added the word "and" in the sentence: “y have added the barrs — and in our study —
- Sensitisation was incorrect (British spelling); it has been changed to sensitization (correct American spelling).
- Recognised was incorrect (British spelling); it has been changed to recognized (correct American spelling).
- I replaced "In general" with "Overall" for clarity.
I believe these changes improve the accuracy and clarity of the manuscript. Thank you for your consideration.
Kind regards,
M. Soledad Zamarro Parra

Reviewer 2 Report (Previous Reviewer 1)
Comments and Suggestions for Authors
The article refers to a preliminary analysis of lemon, orange, and tangerine germin allergens. The rationale and methodology used in the article are adequate. The results are well presented in the figures, even though there are four patients. I would suggest modifying the title of the manuscript to include preliminary report. The discussion and conclusions are informative. It would probably be essential to add possible cross-reactivity with other citric fruits. Finally, the authors should add information on conflicting interests, financial support, and the role of each one in the manuscript
Author Response
1. We agree with the reviewer that the preliminary nature of the study should be emphasized. The manuscript title has been updated accordingly: Title modification:
"Orange Allergy Beyond LTP: IgE Recognition of Germin-like Proteins in Citrus Fruits"
-
We appreciate the reviewer’s suggestion and have expanded the Discussion section to address potential cross-reactivity. Although oral food challenges were not performed in all patients, the presence of homologous germin-like proteins in orange, lemon, and mandarin suggests possible IgE cross-reactivity, which we highlight as a relevant hypothesis for future studies.
We now state (Discussion, lines-----453-455):
The identification of germin-like proteins in orange, lemon peel, and mandarin peel suggests possible IgE cross-reactivity among citrus fruits. Although not directly assessed in this study, this phenomenon may partially explain sensitization patterns observed in patients with orange allergy and warrants further investigation
Conflict of Interest, Funding, and Author Contributions
In accordance with the journal's policies and the reviewer’s comment, we have included the following statements at the end of the manuscript:
-
Conflict of Interest:
“The authors declare no conflicts of interest.”
-
Funding Sources:
“This research received no external funding.”
-
Author Contributions:
“MSZ: clinical evaluation of patients, conceptualization, manuscript writing. MMG, AH, JA, ID, DR, RP: laboratory procedures. AC: supervision and data interpretation. All authors reviewed and approved the final version of the manuscript.”

Reviewer 3 Report (New Reviewer)
Comments and Suggestions for Authors
Authors wished to characterize the sensitization pattern in five patients who reported allergic reactions following orange ingestion. They did not find any new allergens, but they just observed IgE reactive with various molecular weight, which had not been furtherly characterized. The only allergen they found is the well known Cit s1 (germin).
More information about allergic reaction of the two patients who reported anaphylaxis is needed.
Authors should report whether their patients reported allergic reactions with lemon and mandarin and, in case they did not, Authors should explain the reason.
It is curious that the patient n.5 who reported oral allergic syndrome was not sensitized to profilin.
Author Response
“They did not find any new allergens, but they just observed IgE reactive with various molecular weight, which had not been furtherly characterized. The only allergen they found is the well known Cit s1 (germin).”
We thank the reviewer for this observation. We agree that Cit s 1 (germin) is already annotated as a known allergen in orange. However, our study provides the first identification of germin-like proteins in lemon and mandarin peel, confirmed by peptide fingerprinting and IgE binding. This finding expands the allergen repertoire in these citrus fruits and has not been previously reported in the IUIS database.
We have revised the Introduction and Objective to clarify that the aim was not to discover novel allergens, but to characterize the presence and IgE recognition of non-LTP proteins, particularly germin, in citrus fruits other than orange.
“More information about allergic reaction of the two patients who reported anaphylaxis is needed.”
We have added detailed information on the anaphylactic episodes of the two patients (Patients 1 and 2) in Table 3 and in the Results section. Both reactions occurred immediately after ingestion of orange and were classified as oFASS grade 4. Both patients required emergency medical attention, and epinephrine was administered.
We now state in the Results:
“Patient 1 developed generalized urticaria, facial angioedema, and respiratory symptoms within minutes of consuming orange without adding coafactors. Adrenaline was administered at the ER (emergency room). Patient 2 reported a similar episode with cardiovascular symptoms and also required epinephrine at her home without apparent cofactors, only stress reported by the patient.”
“Authors should report whether their patients reported allergic reactions with lemon and mandarin and, in case they did not, Authors should explain the reason.”
We thank the reviewer for pointing this out. Among the five patients, only Patients 1 and 5 underwent oral challenge with lemon and mandarin, both of which were well tolerated without peel. The remaining patients had no history of clinical reaction to lemon or mandarin, and were not challenged for ethical reasons, as they had experienced severe reactions to orange and they refused to oral challenge. They already tolerated lemon in juice.
We now clarify this in the Methods and Results sections line 289:
“Oral food challenges with lemon and mandarin were performed in Patients 1 and 5, with no adverse reactions observed. In the remaining cases, no clinical history of reactions to these fruits was reported, and they all tolerated in lemon juice, they refused to eat it whole.”

Round 2
Reviewer 3 Report (New Reviewer)
Comments and Suggestions for Authors
The revised manuscript has encorporated all the criticisms and suggestions and it is mauch improved.
This manuscript is a resubmission of an earlier submission. The following is a list of the peer review reports and author responses from that submission.
Round 1
Reviewer 1 Report
Comments and Suggestions for Authors
The communication, too short for an article, refers to germin purification from lemon, orange, and tangerine and its relation to the allergic response. Even though the communication may be relevant from the protein context, which was not thoroughly explored, sequence, solubility, pI, etc., the manuscript contains only the analysis of 5 samples. From Figure 1, it is unclear if the proteins are associated with other proteins or may be degraded. Figure 2, the recognition of two bands in tangerine peel suggests cross-binding? I am unsure of the results of patient 4. In addition, why do the authors did not do the specific immunoprecipitation using the classic IgG anti-IgE and protein A sepharose? There are details to correct in Figure 2, such as Spanish words. Is it possible that those patients had IgG against the proteins? Finally, what is the future prespective? Why did the authors did not cite important manuscripts as this one? DOI: 10.1034/j.1398-9995.2002.23686.x
The manuscript needs to be rewritten probably more experiments, using a higher number of negative samples would enhance the manuscript
Reviewer 2 Report
Comments and Suggestions for Authors
Review of the Article:
Germin: relevant Allergen in lemon, citron and tangerine
The manuscript under review explores an important scientific topic: identifying allergenic fractions in citrus fruits, specifically lemon, citron, and tangerine. Although the results are promising, the manuscript in its current form requires substantial additions, clarifications, and major revisions. Due to these deficiencies, it is challenging to assess the scientific value of the study. Additionally, the language used is often unclear, with frequent use of incomplete sentences, making the text difficult to follow.
Specific Comments and Suggestions:
Abstract: The abstract should be rephrased, especially in terms of describing the methodology. Additionally, the allergen nomenclature is incorrect: "Cit" allergens should be used consistently named instead of “Cyt.”
Introduction: This section requires expansion. Some background information on the symptoms associated with allergies to orange, citron, and tangerine should be included. It is important in the context of analysed symptoms of patients involved to the study. Data of allergy prevalence should be based on primary sources rather than secondary ones that only reference the data indirectly (f.ex. results from the EuroPreval Community surveys). A more structured and comprehensive overview of the topic would strengthen the foundation of the study. The aim of the study should be clearly stated. In this form it is not clear if it was case study or analytical study.
Materials and Methods:
This section requires significant reorganisation and completion, as several critical methodological details are missing.
Specific suggestions include:
1. Providing a more detailed description of the "Prick-Prick" test methodology.
2. Clarifying the term “Bjorksten Liquid Tamon”
3.In the SDS-PAGE procedure: details about gel size including producer, the quantity of protein applied per lane (the concentration of the extract is not the same), electrophoretic protein separation parameters, and the molecular weight marker (name of producer) should be included, the software and procedure used for determining molecular weights should also be specified.
Western Blotting: the description of the Western blot method and the parameters used for protein transfer from gel to membrane requires additions and clarifications, the immunostaining procedure is ambiguous and needs clarification. Which substrate was used 4CN or ECL, both one after one?! The abbreviations require explanation. Some necessary information on the secondary antibodies, including manufacturer and whether polyclonal or monoclonal antibodies were used, are required.
Above are just only examples, overall, this section requires thorough review and refinement to ensure all methodological details are complete.
Results: Patient characteristics should be presented in a table for clarity, as the current format is difficult to interpret. Figure captions are insufficient and need rephrasing to accurately reflect the results presented.
Discussion: The discussion section is overly brief and lacks depth. A more thorough analysis is needed to place the findings in context with existing scientific literature.
In the summary, this manuscript requires extensive revision and reassessment. Enhancing the clarity, completeness, and coherence of the text is essential to effectively communicate the scientific findings.
Comments on the Quality of English Language
The language used is often unclear, with frequent use of incomplete sentences, making the text difficult to follow, f.ex. - see material and methods description in abstract, in Result section description of patients